# Role of Angiopoietins in Development of Cancer and Neoplasia Associated with Viral Infection

**DOI:** 10.3390/cells9020457

**Published:** 2020-02-18

**Authors:** Xiaolan Yu, Fengchun Ye

**Affiliations:** 1State Key Laboratory of Biocatalysis and Enzyme Engineering, College of Life Sciences, Hubei University, Wuhan 430062, China; 2Hubei Key Laboratory of Industrial Biotechnology, College of Life Sciences, Hubei University, Wuhan 430062, China; 3Department of Molecular Biology & Microbiology, School of Medicine, Case Western Reserve University, 10900 Euclid Avenue, Cleveland, OH 44106, USA

**Keywords:** angiopoietin-1 (Ang-1), angiopoietin-2 (Ang-2), angiogenesis, cancer, neoplasia, oncogenic virus

## Abstract

Angiopoietin/tyrosine protein kinase receptor Tie-2 signaling in endothelial cells plays an essential role in angiogenesis and wound healing. Angiopoietin-1 (Ang-1) is crucial for blood vessel maturation while angiopoietin-2 (Ang-2), in collaboration with vascular endothelial growth factor (VEGF), initiates angiogenesis by destabilizing existing blood vessels. In healthy people, the Ang-1 level is sustained while Ang-2 expression is restricted. In cancer patients, Ang-2 level is elevated, which correlates with poor prognosis. Ang-2 not only drives tumor angiogenesis but also attracts infiltration of myeloid cells. The latter rapidly differentiate into tumor stromal cells that foster tumor angiogenesis and progression, and weaken the host’s anti-tumor immunity. Moreover, through integrin signaling, Ang-2 induces expression of matrix metallopeptidases (MMPs) to promote tumor cell invasion and metastasis. Many oncogenic viruses induce expression of Ang-2 to promote development of neoplasia associated with viral infection. Multiple Ang-2 inhibitors exhibit remarkable anti-tumor activities, further highlighting the importance of Ang-2 in cancer development.

## 1. Introduction

A hallmark breakthrough in vascular biology during the 1990s was the discovery of angiopoietins, ligands of the tyrosine kinase receptor Tie-2 [1,2,3,4,5]. Angiopoietin-1 (Ang-1) and angiopoietin-4 (Ang-4) act as agonists of Tie-2 whereas angiopoietin-2 (Ang-2) and angiopoietin-3 (Ang-3) act as antagonists [1,2,3,4,5,6]. Extensive studies have revealed their essential roles in angiogenesis and wound healing through Tie-2 signaling, with most of the studies focusing on Ang-1 and Ang-2. The roles of Ang-3 and Ang-4 are far less understood. Ang-1 was found to be essential for the maturation and “sealing” of newly formed blood vessels [1,7,8]. Blood vessel pericytes strongly express Ang-1 and were found to be recruited to maturing micro-vessels during later stages of cutaneous wound healing, leading to their sealing and maturation [8,9]. Ang-1 deficient mice died early of hemorrhage as a result of generating “unsealed” and “leaking” blood vessels [4,10]. In contrast, Ang-2 displays characteristic features of an antagonist of Tie-2. Mice over-expressing Ang-2 manifested hemorrhage [3], most likely by antagonizing Ang-1. In collaboration with vascular endothelial growth factor (VEGF), Ang-2 was found to play a crucial role in the initiation of angiogenesis by destabilizing existing blood vessels for the generation of new blood vessels [1,11]. Besides Ang-2/Tie-2 signaling, a recent study demonstrated that Ang-2 destabilization of existing blood vessels also depended on Ang-2-mediated activation of integrin-β1 [12]. Consistent with this role, Ang-2-deficient mice died early due to failure of angiogenesis [3,13]. Up to now, a wealth of studies have firmly confirmed the opposing but complementary effects of Ang-1 and Ang-2 on angiogenesis and wound healing. For a review of these studies, please refer to a number of comprehensive review articles published elsewhere [14,15,16].

During the past two decades, the roles of angiopoietins have extended far beyond angiogenesis and wound healing. It is now clear that these molecules contribute to multiple other aspects of biology such as inflammation [17], cell survival [18], and cell migration and invasion [19]. Indeed, dysregulation of these molecules has been associated with a number of diseases including infection and septic shock [20,21], diabetes [22,23], and cancer [24,25]. In this article, we attempt to give an update reviewing recent literature on how Ang-1 and Ang-2 contribute to development and progression of cancer in general and neoplasia associated with viral infection.

## 2. Angiopoietins and Cancer

### 2.1. Dysregulation of Angiopoietins in Cancer

In healthy people, the level of Ang-1 in circulation is relatively high, which is likely necessary for stable maintenance of the integrity of existing blood vessels [7,26]. In contrast, expression of Ang-2 is limited, which is consistent with low levels of angiogenesis in healthy individuals [26,27]. In cancer patients, however, this expressional pattern of Ang-1 and Ang-2 is perturbed. The serum levels of Ang-2 in cancer patients increase and the ratio between Ang-1 and Ang-2 in circulation decreases significantly [28,29,30,31]. This alteration in Ang-1 and Ang-2 expressional patterns in cancer patients seems to be concordant with the well-defined functions of these two angiogenic factors described earlier. Indeed, tumor blood vessels are considered abnormal when compared to blood vessels in normal tissues. Tumor vessels are tortuous and leaky, their diameter is irregular and their walls are thin [32,33,34]. A relative deficiency of pericytes could be responsible for these morphological features in tumor vasculature as a result of the altered expression pattern of Ang-1 and Ang-2 in cancer patients [32,33,34]. A number of clinical studies have demonstrated a strong inverse correlation between the serum levels of Ang-1 and Ang-2 and prognosis of cancer [28,35,36,37,38,39,40], suggesting important roles of these molecules in cancer development and progression.

The mechanisms of Ang-2 up-regulation in cancer patients have been investigated quite extensively. Endothelial cells are the main source of Ang-2, expression of which is restricted to very low levels in healthy people. The promoter of Ang-2 contains both positive and negative cis-elements for transcriptional activation and repression [41]. The E26 transformation-specific (Ets) family transcription factors Ets-1 and Elf-1 and other transcription factors such as the activating protein 1 (AP-1) and forkhead box protein C2 (FOXC-2) act as positive regulators or trans-elements [42,43,44]. The Ang-2 gene promoter contains multiple Ets-1 and Elf-1 binding sites (cis-elements) for cytokine-dependent transcriptional induction [45]. The negative regulatory trans-elements remain unknown. However, the DNA of Ang-2 promoter is highly methylated, which inversely correlates with Ang-2 expression in cancer cells [46], suggesting that Ang-2 promoter is subject to epigenetic repression.

Hypoxia and cytokines have been found to be the main triggers of Ang-2 up-regulation in cancer patients. Hypoxia contributes a great deal to solid tumor progression and development of therapy resistance [47], and cancer cells are known to produce various growth factors and cytokines [48]. Both hypoxia and cytokines were shown to be strong inducers of Ang-2 expression in cultured endothelial cells [49,50,51,52,53] Hypoxia and cytokines are known to activate transcription factors such as Ets-1 involved in Ang-2 transcriptional induction [54,55]. Among various cytokines, tumor-derived VEGF was found to be a potent inducer of Ang-2 expression in host endothelium, which destabilized host vasculature and promoted angiogenesis in ovarian cancer [56]. The inflammatory mediator oncostatin M (OSM) also induced Ang-2 expression in endothelial cells both in vitro and in vivo [57]. In addition, the female hormone estrogen was suggested to increase Ang-2 expression in female rats [58]. Besides induction of Ang-2 expression, cytokines also stimulate release of Ang-2 that is pre-made and stored in the Weibel–Palade bodies (WPB) of endothelial cells. For instance, upon stimulation by factors such as thrombin, Ang-2 was rapidly released from endothelial cells [59,60,61].

### 2.2. Role of Angiopoietins in Tumor Angiogenesis and Tumor Growth

Angiogenesis is essential for growth of solid tumors and metastasis [62]. By using an immune deficient mouse model with implanted human lung cancer cells, Holopainen et al. demonstrated that mice treated with adenoviruses expressing Ang-1 did not give rise to tumors with significantly increased vascular density [63]. Rather, this treatment enlarged blood vessels in both the tumor and normal tissues, increased tumor cell dissemination into the blood circulation [64,65], and enhanced the formation of metastatic foci in the lungs. Simultaneous treatment of the mice with soluble Tie2 attenuated this effect of Ang-1. These results suggest that the Ang-1/Tie-2 signaling increases vascular entry and exit of tumor cells to facilitate tumor dissemination and metastases by giving rise to enlarged blood vessels. In contrast to this study, Ang-1 did enhance tumor angiogenesis in a rat glioma model [66]. In addition, multiple studies demonstrated that Ang-1 actually inhibited tumor growth [67,68]. One study found no detectable effect of Ang-1 on dissemination of Lewis lung carcinoma and TA3 mammary carcinoma cells [69]. Therefore, the role of Ang-1 in tumor angiogenesis and growth seems to be highly dependent on the specific type of cancer.

In contrast to Ang-1, Ang-2 overexpression gave rise to aberrant “leaky” blood vessels or aggregated vascular endothelial cells with few associated smooth muscle cells, suggesting that Ang-2 does play a role in regulating tumor angiogenesis [69]. Similar results were also obtained with other cancer models including pancreatic cancer, liver cancer, breast cancer, as well as colon cancer [70,71,72,73]. In addition to Ang-2/Tie-2 signaling, Ang-2 may promote tumor angiogenesis through additional mechanisms. Immuno-histochemical analysis data showed that Ang-2 in human gastric cancer biopsies was predominantly localized in cancer tissues when compared with normal tissues, and was expressed not only in endothelial cells but also in cancer cells [74]. Nude mice implanted with tumor cells over-expressing Ang-2 into the gastric walls developed highly metastatic tumors with hypervascularity as compared with mice implanted with tumor cells expressing the control vector. In addition, there was a significant correlation between Ang-2 expression and lower grade vessel maturation. Furthermore, higher levels of proteases such as matrix metalloproteinase-1 (MMP-1), matrix metalloproteinase-9 (MMP-9), and urokinase-type plasminogen activator were detected in the tumor tissues than in normal tissues, suggesting possible involvement of these proteases in Ang-2 mediated tumor angiogenesis. In full agreement with this notion, all three proteases in endothelial cells were up-regulated upon treatment with Ang-2 in the presence of VEGF in vitro, further suggesting that Ang-2 contributes to tumor angiogenesis by induction of proteases in endothelial cells [74]. The roles of Ang-3 and Ang-4 in cancer development remain largely undefined. Only one study reported that Ang-3 inhibits pulmonary metastasis by inhibiting tumor angiogenesis [75], demonstrating inhibition of endothelial cell proliferation and thus reduced angiogenesis when Ang-3 is over-expressed in the tumor cells.

In brief, despite some contradictory findings in different tumor models, most studies have found critical roles of both Ang-1 and Ang-2 in tumor angiogenesis [76,77,78,79]. As summarized in Figure 1, due to constant higher levels of Ang-2 in cancer patients, tumor angiogenesis is a continuous process that supports tumor growth and progression, which is in contrast to the transient feature of angiogenesis during wound healing. In addition, as a result of reduced Ang-1 to Ang-2 ratio, tumor angiogenesis often generates unsealed “leaky” blood vessels. The role of Ang-1 and Ang-2 in tumor angiogenesis is further supported by recent studies showing remarkable effectiveness of several Ang-1/Ang-2 specific inhibitors and neutralizing antibody in blocking tumor angiogenesis and tumor growth [64,65,80,81,82].

### 2.3. Roles of Angiopoietins in Tumor Invasion and Metastasis

Accumulating evidence points to a tight association between Ang-2 expression and tumor invasion and metastasis in various human cancers [78,83,84,85], which apparently goes far beyond the angiogenic effect of Ang-2. Indeed, Ang-2 stimulated invasion of glioma and breast cancer cells through up-regulation and activation of matrix metalloprotease 2 (MMP-2) in the tumor cells [19,72,86]. Although Ang-2 is a specific ligand of Tie-2, apparently it also interacts with other receptors. Indeed, Ang2 interacted with α5β1 integrin in Tie2- deficient human glioma cells, leading to activation of focal adhesion kinase (FAK), p130Cas, extracellular signal-regulated protein kinase (ERK) 1/2, and c-jun NH2-terminal kinase (JNK) and induction of MMP-2 expression and secretion [19,72,86]. Expressional knocking down (KD) of Ang-2 resulted in simultaneous reduction of MMP-2 expression and metastasis of human pancreatic carcinoma [87]. Ang-2 may induce additional proteinases to promote tumor cell invasion and metastasis [88].

In addition to Ang-2, high levels of Ang-1 have also been associated with capsular invasion, extrathyroid extension, lymphovascular invasion, lymph node metastasis, and recurrence in patients [89], although the underlying mechanisms remain obscure.

### 2.4. Roles of Angiopoietins in Tumor Inflammation and Microenvironment

Angiopoietins, particularly Ang-2, also play important roles in inflammation. One study demonstrated that administration of Ang-2, but not Ang-1, induced edema in the mouse paw in a dose-dependent manner [90], which was blocked by co-administration of soluble Tie-2. Both Ang-1 and Ang-2 demonstrated the abilities to attract migration of inflammatory cells such as neutrophils and monocytes [90,91,92,93,94]. Chemotaxis seems to be a driving force for Ang-1/Ang-2 mediated migration of inflammatory cells [92]. However, the angiopoietin/Tie-2 signaling most likely is involved in this event as well. Indeed, significant numbers of neutrophils and monocytes express weak to moderate levels of the Tie-2 receptor, and the effects of both Ang-1 and Ang-2 on neutrophil/monocyte migration were shown to be Tie-2 dependent [91,95,96]. This event may engage additional mechanisms as well. In fact, Ang-2 also promoted monocyte infiltration in a β2-integrin-dependent manner [97,98].

Tumor-associated macrophages (TAMs) are important drivers of tumor angiogenesis [99]. The Tie-2 expressing monocytes (TEMs) are present in blood and rapidly differentiate and polarize into TAMs once they are recruited into tumors [100]. Elimination of TEMs in various tumor models suppressed tumor angiogenesis [101,102]. Global gene expression analysis indicated that circulating TEMs were already preprogrammed in the circulation to be more angiogenic and expressed higher levels of such proangiogenic genes as MMP-9, vascular endothelial cell growth factor A (VEGFA), cyclooxygenase-2 (COX-2), and Wnt family member 5A (WNT5A) than Tie2-negative monocytes [103]. Ang-2, produced by tumor blood vessel endothelial cells, not only attracted TEMs to migrate into tumors but also had an impact on global gene expression of TEMs through both Tie-2 and integrin signaling [98,101,104]. Ang-2 enhanced the proangiogenic activity of TEMs and increased their expression of two proangiogenic enzymes: thymidine phosphorylase (TP) and cathepsin B (CTSB). In another study, Ang-2 also increased expression of interleukin-10 (IL-10), mannose receptor (MRC-1), and chemokine (C-C motif) ligand 17 (CCL-17) in TEMs, which are three markers for the so-called “pro-tumor M2-like macrophages [103]. Consistent with the gene expression profile of TEMs, tumors grown in transgenic mice with Ang-2 overexpression specifically by endothelial cells were significantly more vascularized and contained greater numbers of TEMs than tumors grown in wild-type mice [103].

Besides promoting vascular angiogenesis, the Ang-2/TEMs interaction contributes to tumor lymphangiogenesis as well. In malignant tumors of untreated breast cancer patients, TEMs expressed the canonical lymphatic markers LYVE-1, Podoplanin, VEGFR-3 and PROX-1 [105]. In addition to acquisition of these lymphatic markers, TEMs inserted into lymphatic vessels in tumors but not in the adjacent non-neoplastic tissues, suggesting that the tumor microenvironment shapes both TEM phenotype and spatial distribution [105].

The interaction between Ang-2 and TEMs may regulate the tumor microenvironment in additional ways, including weakening of the host’s anti-tumor immunity. Ang-2 was found to markedly inhibit release of the important anti-tumor cytokine, tumor necrosis factor-alpha (TNF-α), by monocytes in vitro [106]. Following extravasation of monocytes and their differentiation into macrophages, many TEMs accumulated in the hypoxic areas of inflamed and malignant tissues. As mentioned earlier, hypoxia up-regulates Ang-2 expression in endothelial cells and Tie-2 expression in monocytes and macrophages. Hypoxia also augmented the inhibitory effect of Ang-2 on the release of the anti-angiogenic cytokine, interleukin-12 (IL-12) by monocytes [106]. Moreover, Ang-2 not only augmented expression but also stimulated release of the potent immunosuppressive cytokine, IL-10, from TEMs, which is a chemokine for regulatory T cells (T-reg). IL-10 suppressed T cell proliferation, increased the ratio of CD-4+ T cells to CD-8+ T cells, and promoted the expansion of CD-4+CD-25(high)FOXP-3+ T-reg [107]. Accordingly, syngeneic murine tumors expressing high levels of Ang-2 contained not only high numbers of TEMs but also increased numbers of T-reg, whereas genetic depletion of tumor TEMs resulted in a marked reduction in the frequency of T-reg in tumors. Therefore, the Ang-2/TEMs interaction axis represents a potent immunosuppressive force in tumors.

In summary, as shown in Figure 2, Ang-2 contributes to cancer development in multiple ways. Through Tie-2 signaling and in collaboration with VEGF, Ang-2 promotes tumor angiogenesis. Through both Tie-2 and integrin signaling, Ang-2 induces chemotaxis and migration of myeloid cells including TEMs into tumors, which subsequently become tumor-promoting TAMs. Through integrin signaling, Ang-2 also induces expression of MMPs to promote tumor invasion.

## 3. Angiopoietins and Neoplasia Associated with Viral Infection

Infection accounts for over 15% of all cancers [108], with viral infection being a leading cause of infection-associated cancers [109,110,111]. A number of viruses have been identified as oncogenic viruses that can directly transform normal cells into malignant tumor cells. Examples of such oncogenic viruses include: (1) Epstein–Barr virus (EBV), which is associated with a subset of Hodgkin’s lymphoma [112,113,114,115,116,117], a subset of diffuse large B-cell lymphoma (DLBCL) [118], endemic Burkitt lymphoma [119], as well as nasopharyngeal carcinoma [119,120] and gastric adenocarcinoma [121]; (2) Kaposi’s sarcoma-associated herpesvirus (KSHV), which is associated with Kaposi’s sarcoma (KS) [122], primary effusion lymphoma (PEL) [123,124], and multicentric Castleman disease (MCD) [125]; (3) High-risk isotypes of human papillomaviruses (HPV), which can cause cervical, anal, oral, vulvar, vaginal, and penile cancers [126,127,128,129,130]; (4) Merkel cell polyomavirus (MCV), which is linked to Merkel cell carcinoma (MCC) [131]; and (5) human T-cell leukemia virus type 1 (HTLV-1) [132,133,134,135,136]. Other viruses such as hepatitis B virus (HBV) and hepatitis C virus (HCV) can indirectly cause liver cancer through chronic inflammation [137]. A unique feature of virus infection-associated cancers is that most oncogenic viruses directly induce expression of angiogenic and inflammatory cytokines including angiopoietins to promote tumor angiogenesis, inflammation, and tumor growth [138,139,140,141]. As an example, in the following sections, we will review the literature on how KSHV induces expression of Ang-2 to promote development of KS, which is the most common malignancy in people infected with human immune deficiency virus (HIV) [142].

### 3.1. Ang-2, But Not Ang-1, Is Highly Expressed in KS Tumors

Kaposi’s sarcoma (KS) is a neoplasia of endothelial cell origin [143,144]. Early-stage KS is not a true malignancy but hyperplasia driven by KSHV-induced angiogenesis, inflammation, and proliferation [144,145,146]. Due to accumulative genetic mutations over time, the hyperplasia further progresses into malignant tumors. KS lesions consist of not only the proliferative KSHV-infected tumor cells that express endothelial cells-specific markers but also a large number of infiltrating inflammatory cells including lymphocytes, monocytes and macrophages [147]. Another feature of KS lesions is the extensive “slit-like” leaky blood vessels [144,148,149,150]. It is believed that inflammation and dysregulated angiogenesis play a crucial role in the development and progression of KS.

Brown et al. conducted the first study that examined possible involvement of Tie-2 and angiopoietins in KS [151]. High levels of Ang-2, Tie-1, and Tie-2 mRNAs were detected in KS tumors. However, no Ang-1 mRNA was detected. Consistent with this report, two later studies demonstrated strong expression of Ang-2 protein in KS biopsies [44,152]. KSHV infection was held responsible for the elevated expression of Ang-2 and other cytokines in KS [153,154,155,156]. First, KSHV infection of endothelial cells strongly induced Ang-2 transcription via activation of Ets-1 and AP-1 [44,154], two crucial transcription factors involved in Ang-2 transcription, which was mediated by the mitogen-activated protein kinase (MAPK) pathway [41,45]. Inhibition of these signaling pathways attenuated KSHV-induced Ang-2 transcription. Two KSHV-encoded cytokines named viral interleukin-6 (vIL-6) and viral G-protein–coupled receptor (vGPCR) were found to induce Ang-2 transcription in lymphatic endothelial cells through activation of the MAPK pathway as well [155]. These viral proteins are expressed at the highest levels about two days after KSHV infection [157]. Indeed, strongest level of Ang-2 mRNA was detected at 54 h post-infection [44], further supporting the involvement of vIL-6 and vGPCR in Ang-2 transcriptional induction. Interestingly, HBV and HCV also induced Ang-2 expression through activation of the MAPK pathways [138]. However, these two viruses may engage host cytokines to induce Ang-2 expression as they do not encode any viral cytokines. Additional regulatory mechanisms and viral proteins for KSHV-induced Ang-2 transcription may exist as well. For instance, Nutlin-3, which activates the tumor repressor p53 [158], repressed Ang-2 expression in KSHV-infected endothelial cells [159], which may be indicative of a role of p53 in repressing Ang-2 expression. This tumor repressor is known to repress transcription of VEGF [160]. It is important to note that p53 is one of the most important tumor suppressor genes [161], which is either lost or mutated in about 50% of all cancers [162,163]. Many oncogenic viruses including KSHV inactivate p53 [164,165,166]. Nevertheless, two studies did not find a significant association between p53 and Ang-2 expression in other types of cancer [167,168].

In addition to induction of Ang-2 transcription, KSHV also was also found to stimulate immediate release of Ang-2 that is pre-synthesized and stored in the Weibel–Palade bodies (WPB) of endothelial cells [156]. KSHV binding to integrin receptors triggered tyrosine phosphorylation of the focal adhesion kinase (FAK), the tyrosine kinase Src, and the Calα2 subunit of the l-type calcium channel. These sequential events led to rapid calcium (Ca^2+^) influx and Ang-2 release, which could be blocked by pretreatment of endothelial cells with protein tyrosine kinases inhibitors and calcium channel blockers.

### 3.2. Ang-2 Promotes Angiogenesis, Inflammation, and KS Tumor Growth

Initially, in a “tumor cell free” angiogenesis assay, Ye et al. demonstrated that Matrigel blocks filled with supernatant from KSHV-infected human umbilical vein endothelial cells (HUVECs) displayed significantly higher numbers of micro-vessels than those filled with supernatants from mock-infected HUVECs two weeks after inoculation into mice [44]. Adding Ang-2 neutralization antibody into the Matrigel effectively blocked the angiogenic effect of KSHV-induced Ang-2. This data is concordant with a recent study using a KSHV-induced endothelial cell tumor model that strongly expressed Ang-2 [94,169]. When equal numbers of the KSHV-induced tumor cells were used in a Matrigel-based angiogenesis assay in nude mice, inclusion of AMG-138 or L1-10, two peptide-based Ang-2 inhibitors [81], substantially reduced the numbers of blood vessels. Moreover, shRNA KD of Ang-2 expression in these tumor cells had similar effects. Therefore, Ang-2 is a crucial for KS tumor angiogenesis.

In addition to blocking tumor angiogenesis, the two Ang-2 inhibitors also significantly inhibited KSHV-induced tumor growth in nude mice [94]. Consistent with this result, Ang-2 KD in the tumor cells also led to reduced tumor sizes. Furthermore, there were significantly fewer numbers of infiltrating mouse CD-16 positive myeloid cells in the tumors from mice treated with the two inhibitors or inoculated with Ang-2 KD tumor cells [94]. This result indicated that Ang-2 played a key role in attracting monocytes/macrophages to infiltrate into tumors. Indeed, in an in-vitro migration assay using immortalized human monocytes, KSHV-induced Ang-2 strongly enhanced migration of monocytes [94]. Chemotaxis may be a driving force for Ang-2 induced monocyte migration. However, as described earlier, Ang-2 directly interacts with monocytes and TEMs through both Tie-2 and integrin signaling [170,171,172]. These myeloid cells, once present in a KSHV infection microenvironment, rapidly differentiated into TAMs [157,173], which, as described earlier, are well known to promote tumor growth and progression [171,174,175]. Therefore, as shown in Figure 3, KSHV-induced Ang-2 plays a pivotal role in the development of KS by promoting both tumor angiogenesis and inflammation.

## 4. Conclusions and Perspectives

Dysregulation of Ang-1 and Ang-2 is a characteristic feature of cancer patients. Elevated serum levels of Ang-2 inversely correlate with cancer prognosis. Hypoxia, cytokines, and infection by oncogenic viruses induce Ang-2 expression. Ang-2 not only acts on endothelial cells through Tie-2 signaling to promote tumor angiogenesis but also interacts with tumor cells through integrin signaling, leading to elevated expression of MMPs and tumor cell invasion. Ang-2 also interacts with various myeloid cells particularly TEMs through Tie-2 and/or integrin signaling, attracting their migration and infiltration into tumors. These tumor stromal cells secrete various cytokines and other factors to further stimulate tumor angiogenesis, invasion, and metastasis, and inhibit the host’s anti-tumor immunity. Since Ang-2 contributes to cancer development in so many different ways, it has become an important target for chemotherapy. Multiple strategies, including soluble Tie-2, humanized Ang-2 neutralization antibody, and small peptides have achieved remarkable anti-tumor effectiveness in various cancer models. These Ang-1/Ang-2 blocking molecules definitely offer a new promising treatment for cancer including neoplasia associated with viral infection.

## Figures and Tables

**Figure 1 cells-09-00457-f001:**
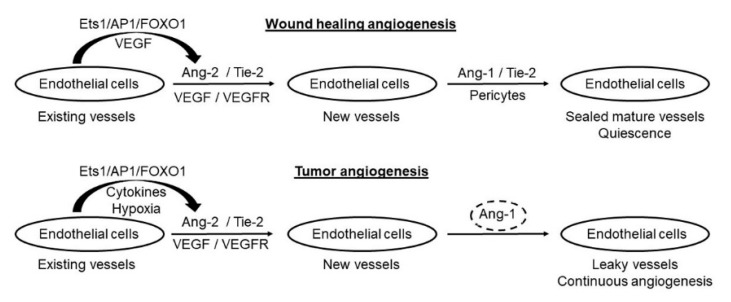
Differences between wound healing angiogenesis and tumor angiogenesis. With constant production of high levels of Ang-2, tumor angiogenesis is a continuous process that supports tumor growth and progression. In contrast, Ang-2 is only transiently induced during injury, and angiogenesis is halted right after wound healing. Secondly, high levels of Ang-1 assures maturation and “sealing” of newly generated blood vessels by recruiting pericytes during wound healing. However, due to insufficient levels of Ang-1 in cancer patients, tumor angiogenesis generates unsealed “leaky” blood vessels.

**Figure 2 cells-09-00457-f002:**
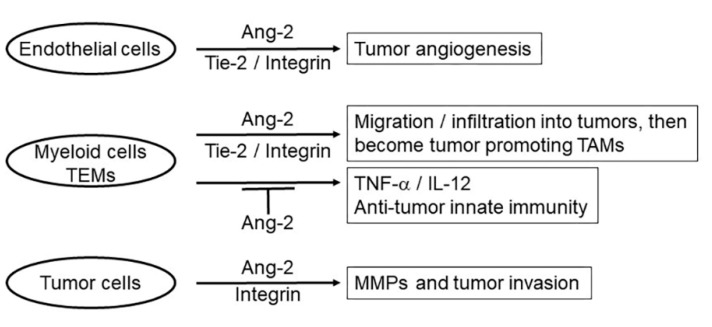
Ang-2 contributes to cancer development through multiple mechanisms. Acting on endothelial cells through Tie-2/integrin signaling, Ang-2 promotes tumor angiogenesis. Acting on myeloid cells including TEMs through Tie-2 and integrin signaling, Ang-2 induces chemotaxis and their migration into tumors, which subsequently become tumor-promoting TAMs. Ang-2 also blocks production of TNF-α and IL-12 from the myeloid cells to undermine their anti-tumor activity. Ang-2 can also act on tumor cells through integrin signaling, inducing expression of MMPs to promote tumor invasion.

**Figure 3 cells-09-00457-f003:**
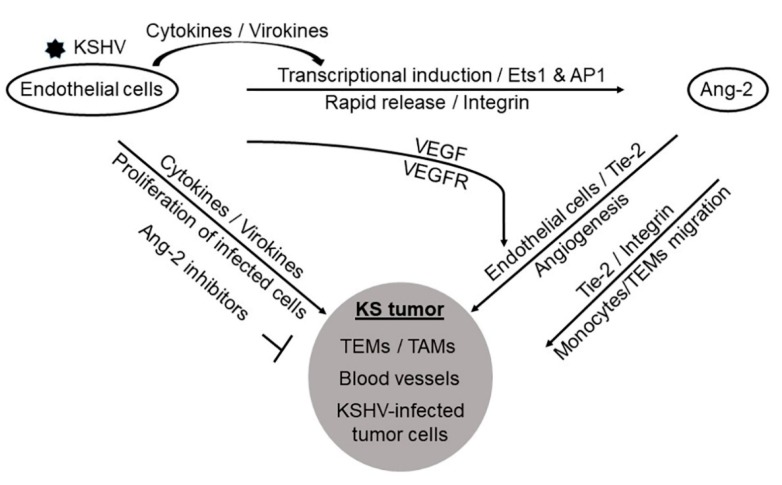
KSHV-induced Ang-2 promotes angiogenesis, inflammation, and KS tumor development. KSHV acute infection of endothelial cells strongly induces Ang-2 transcription via activation of the transcription factors Ets1 and AP1, and triggers its rapid release through integrin signaling. The KSHV-induced Ang-2 acts on uninfected endothelial cells to promote angiogenesis in collaboration with KSHV-induced VEGF, and stimulates chemotaxis and migration of monocytes and TEMs towards the infection sites. Proliferation of the KSHV-infected endothelial cells, extensive blood vessels, and infiltration of inflammatory cells are hallmarks of KS tumors. The effective inhibition of KS tumor growth in nude mice by Ang-2 inhibitors strongly suggests that Ang-2 plays a pivotal role in KS tumor development.

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
