# Peer review of "Role of Angiopoietins in Development of Cancer and Neoplasia Associated with Viral Infection"

_cells, 2020, doi:10.3390/cells9020457_

Round 1

Reviewer 1 Report

This is an interesting review concerning Angiopoietins signaling in cancer in general and in cancer associated with viral infection.

There are some issues that need to be addressed:                                              

The authors used the term “neoplasia” in the title to describe the associated viral infections. The use of “cancer” and “neoplasia” in the same sentence can be misinterpreted by the readers. Since the MS is focused in malignant neoplasia, to avoid confusion, (at least) the title should be replaced, perhaps by “…cancer and viral infection-associated cancer”.

The authors stress that viral infection may affect angiopoietins signaling. But the headlines in section 3 miss the point. Section 3.1 and 3.2 indicate how Angiopoietins are expressed in Kaposi sarcoma and how they promote angiogenesis, inflammation and these tumors growth. This has already been reported in section 2 regarding the effect of Ang1/2 in cancer overall! These titles should focus on how the virus proteins affect angiopoietins.

Figure 1 is very confusing and needs to be changed. The effect of Ang1 and Ang2 through Tie2 or integrins, separately in each type of cell would better support the conclusion. In addition, the effect of viral infection and irs involvement in angiopoietins signalling is not straightforward in the Figure. A separate figure for the effect of viral proteins in the angiopoietin signaling pathway would be useful

Author Response

We appreciate the very careful reading of our manuscript and very helpful comments by this reviewer.

1) With regard to comment#1, we still prefer to use the term "neoplasia" for multiple reasons.  First of all, virus-induced tumors start as benign neoplasia. This feature of virus-induced tumors is discussed in Section 3.  For instance, early stage Kaposi's sarcoma (KS) is not malignant at all but a "hyperplasia".  Only late stage KS are malignant neoplasia. However, Ang-2 appears to be vital for both the initiation (benign) and progression (malignant transformation) of KS.  We believe this applies to other virus-induced tumors as well.  In this article, it is our intention to discuss how angiopoietins contribute to development of cancer (malignant) in general and virus-induced tumors at both early and late stages.  Secondly, the term "neoplasia" is commonly used in the tumor virology field.  We therefore decide to make no change to the title as suggested.

2) Section 3 is indeed about how virus infection induces expression of Ang-2 and how the virus-induced Ang-2 contributes to tumor development.  We realize that this section overlaps a little bit with Section 2.  However, this is a review of current literatures with a focus on virus.  We prefer to make no changes to the subtitles either.

3) We truly appreciate the reviewer's critiques and very helpful suggestions on the previous very complicated figure.  We have replaced the previous figure with three new separate figures, each addressing a different point.      

Reviewer 2 Report

## Comments to the authors:

In the manuscript entitled, “Role of Angiopoietins in Development of Cancer and Neoplasia Associated With Viral Infection (cells-699438)”, the authors had a good literature review on the roles and molecular functions of angiopoietins in tumorigenesis and viral infection. Overall, the article was well written and organized. Here are few points the authors may consider to make it more comprehensive.

## Major points:

Since the majority of studies focused on Ang-1/and Ang-2, the authors should also cover the parts of Ang-3 and Ang-4.

The Tie2 signaling is not the only known receptor for angiopoietins. The roles of Tie1 should be integrated to the article.

## Minor points:

Line 109, ‘Ang-1on’ should be ‘Ang-1 on’.

Author Response

We thank the reviewer's positive comments.

For comment/suggestion#1, there is very few literature with regards to how Ang-3 and Ang-4 contribute to cancer.  We only found one paper and added to the text.

For comment/suggestion#2, Tie-1 remains an orphan receptor, and none of the angiopoietins have been shown to act on Tie-1.  Although one old paper suggests that Tie-1 may be a partner of Tie-2, there is no following up studies to support the finding.  We therefore prefer to omit this reference.

The minor typing error has been fixed.

Reviewer 3 Report

The work entitled "Role of Angiopoietins in Development of Cancer and Neoplasia Associated With Viral Infection" presents the summary of information about angiopoietins and their relations to cancer development from the last decade. The review is well written and supported with the references.

Author Response

We appreciate this reviewer's very positive comments.  No issue was mentioned for us to address.